

# Multi-generation Chemical Aging of α-Pinene Ozonolysis Products by Reactions with OH

Ningxin Wang[1], Evangelia Kostenidou[2,3], Neil M. Donahue[1] and Spyros N. Pandis[1,2,3]

[1]Department of Chemical Engineering, Carnegie Mellon University, Pittsburgh

[2]Department of Chemical Engineering, University of Patras, Patra, Greece

[3] Institute of Chemical Engineering Sciences (ICE-HT), FORTH, Patra, Greece

**Abstract**

Secondary organic aerosol (SOA) formation from volatile organic compounds (VOCs) in the atmosphere can be thought of as a succession of oxidation steps. The production of later-generation SOA via continued oxidation of the first-generation products is defined as chemical aging. This study investigates aging in the α-pinene ozonolysis system with hydroxyl radicals (OH) through smog chamber experiments. The first-generation α-pinene ozonolysis products were allowed to react further with OH formed via HONO photolysis. After an equivalent of 2-4 days' of typical atmospheric oxidation conditions, homogeneous OH oxidation of the α-pinene ozonolysis products resulted in a 20-40 % net increase of the SOA for the experimental conditions used in this work. A more oxygenated product distribution was observed after aging based on the increase in aerosol atomic oxygen to carbon ratio (O:C) by up to 0.04. Experiments performed at intermediate relative humidity (RH) of 50 % showed no significant difference in additional SOA formation during aging compared to those performed at low RH of less than 20 %.

## 1. Introduction

Anthropogenic activities such as fuel combustion as well as biogenic sources such as emissions from vegetation can introduce particles and particle precursors into the atmosphere. In most areas, about half of the submicron aerosol mass on average is composed of organic compounds (Zhang et al., 2007). Organic particles directly emitted to the atmosphere are traditionally defined as primary organic aerosol (POA), while those formed through atmospheric





reactions and condensation of species with corresponding volatility are secondary (SOA).
Atmospheric aerosols represent a significant risk to human health by causing respiratory problems
and heart attacks (Davidson et al., 2005; Pope et al., 2009). At the same time these particles
influence the climate of our planet (Intergovermental Panel on Climate Change, 2007).
Oxygenated OA with a high oxygen to carbon ratio (O:C) is often the most important
component of ambient OA suggesting the importance of atmospheric chemistry in the formation
and processing of OA (Zhang et al., 2007). Most laboratory studies of SOA formation so far have
focused on the first stage of reactions involving the target precursor reacting with the chosen
oxidant. In the atmosphere, organic vapors and particles interact with oxidants for days and
therefore successive oxidation processes are inevitable.
Chemical aging refers to the subsequent stages of SOA formation and evolution due to the
production of later-generation products via oxidation of first-generation products by oxidants such
as OH free radicals (Donahue et al., 2006; Henry et al., 2012). Previous studies have explored
various forms of aging, including heterogeneous reactions of oxidants and aerosols (George et al.,
2008), oligomerization (Kalberer et al., 2006), photolysis of either gas or condensed-phase
products (Henry and Donahue, 2012), and homogeneous gas-phase oxidation by OH (Donahue et
al., 2012). Homogeneous gas-phase oxidation reactions appear to be in general much faster than
heterogeneous reactions, due to diffusion limitations of the latter (Lambe et al., 2009). The first-
generation oxidation reactions of most SOA precursors convert much less than 50 % of the
precursor to SOA, leaving more than half of the carbon still in the gas-phase. Additional oxidation
of these vapors can potentially contribute additional and more oxygenated SOA components.
These later-generation reactions have been proposed to be a major missing step connecting
chamber studies to field measurements.
Zeroth order parameterizations have been developed to model the chemical aging of semi-
volatile POA emissions in chemical transport models (Robinson et al., 2007). CTMs using these
schemes show improved performance in urban areas such as Mexico City (Tsimpidi et al., 2011),
but tend to over-predict OA in areas such as the southeastern United States where biogenic VOCs
dominate if chemical aging is assumed to be a major source of additional SOA (Lane et al., 2008).
As a result, the importance of aging of biogenic SOA as a source of SOA mass concentration
remains an issue of debate.





The ozonolysis of α-pinene ($C_{10}H_{16}$) is considered one of the most important global SOA
sources (Griffin et al., 1999). The system has been well characterized through smog chamber
experiments where researchers quantified its SOA yields under different conditions, explored the
reaction pathways and mechanisms, and identified its product distributions. Recent studies suggest
that there is significant potential for additional SOA formation from homogeneous gas-phase aging
by OH of the first-generation α-pinene oxidation products. Major identified products existing in
gas phase such as pinonaldehyde and pinonic acid can serve as SOA precursors and further react
with OH. Pinonaldehyde reacts with OH, with SOA mass yields up to 5 % under low-$NO_x$
conditions and 20 % under high-$NO_x$ conditions (Chacon-Madrid et al., 2013). Müller et al. (2012)
demonstrated the formation of 1,2,3-butanetricarboxylic acid (MBTCA), an SOA product of low
volatility identified in α-pinene ozonolysis, through the gas-phase OH oxidation of pinonic acid.
They reported an experimental yield of 0.6 % for MBTCA from the gas-phase OH oxidation of
pinonic acid, accounting for about 10 % of the total SOA formed. The proposed formation
mechanisms of MBTCA is a classic example of semi-volatile precursors going through oxidation
and forming products of lower volatility.
The Multiple Chamber Aerosol Chemical Aging Study (MUCHACHAS) explored the gas-
phase OH aging effects of the α-pinene ozonolysis products via experiments performed in four
different smog chambers (Donahue et al., 2012). They were able to isolate the aging effect by
using different OH sources (HOOH photolysis, HONO photolysis, TME ozonolysis), light sources
(sunlight, quasi-solar lamps, 350 nm UV lamps), and chambers of different design in size and
material (Teflon and aluminum). Almost in all experiments, additional formation of SOA (up to
55 %) and a more oxidized product distribution (increasing O:C) were observed after aging.
However, in one of the chambers, strong UV photolysis led to decreasing SOA mass
concentrations in experiments with low to moderate OH levels, [OH] $\leq 2 \times 10^6$ molecules $cm^{-3}$
(Henry and Donahue, 2012). These authors concluded that chemical aging involves a complex set
of interacting processes with competing functionalization (conserved C number with products of
lower volatility and higher oxidation states) and fragmentation (cleavage of C-bond with products
over a wide volatility range and higher oxidation states) of the various organic compounds. A 2D-
volatility basis set (2D-VBS) simulation based on these two pathways and a branching ratio
between them showed that homogeneous OH aging can potentially more than double the α-pinene
SOA mass concentration, after about a day's equivalent of typical atmospheric oxidation



conditions. Uncertainties such as "ripening" during which SOA volatility evolves but its mass
remains constant, UV photolysis and heterogeneous OH uptake can further complicate the aging
process.
Qi et al. (2012) also explored aging of the α-pinene ozonolysis system through smog
chamber experiments using HOOH as an OH source and studied the UV photolysis effect. They
observed a 7.5 % increase in the SOA volume concentration and an increase of 0.03 in the O:C
after aging. Minimum photolysis effect was reported for these experiments.
One complication of chamber experiments is the interaction of particles with chamber
walls. The wall-loss rate of particles is a function of particle size, charge distribution, chamber
geometry, turbulence, and electric field within the chamber (Crump and Seinfeld, 1981). In order
to quantify SOA yields from chamber experiments, it is important to correct for particle wall loss.
Recent findings that organic vapors in the chamber can be directly lost to the Teflon walls as well
further complicate the wall-loss correction process (Matsunaga and Ziemann, 2010; Zhang et al.,
2014). Krechmer et al. (2016) measured the loss rate of vapors formed in the chamber and found
the corresponding timescale to be 7-13 min. Ye et al. (2016) determined the vapor wall-loss
timescale in the Carnegie Mellon chamber used in this work to be around 15 min for semi-volatile
organic compounds.
Despite the consensus from the aforementioned chamber studies that gas-phase OH aging
of α-pinene ozonolysis products can contribute to additional SOA formation, there lacks
consistency in the extent to which the additional mass can form for different OH exposures. Part
of the problem is that the estimated amount of additional SOA formed from these long-lasting
aging experiments can be extra sensitive to the particle and the vapor wall-loss correction methods
deployed. The uncertainties at the end of a 10-hour long aging experiment during which most
particles are lost to chamber walls and the measured suspended mass is low can be relatively high.
In this work, we aim to quantify the additional SOA formed during the aging step comparing
measurements from a suite of instrumentation. We adopt a size-dependent particle wall-loss
correction method and develop a procedure to better constrain the associated errors. We also
attempt to constrain the vapor loss using both theoretical calculations and measurements.







## 2. Experimental approach

We conducted experiments in a 12 m$^3$ Teflon (Welch Fluorocarbons) smog chamber at Carnegie Mellon University (CMU). The reactor was suspended in a temperature-controlled room with walls covered with UV lights (GE 10526 and 10244). Prior to each experiment, we flushed the chamber overnight with purified air under UV illumination to remove any residual particles and gas-phase organics. We generated purified air by passing ambient air through a high-efficiency particulate air (HEPA) filter to remove particles, an activated carbon filter to remove any organics, a Purafil filter to remove NO$_x$, and finally a silica gel filter, keeping relative humidity (RH) below 5 % in the chamber before each experiment.

We pumped an ammonium sulfate solution (1 g L$^{-1}$) into the chamber at the beginning of each experiment through an atomizer (TSI, model 3076) at a constant rate of 90 mL h$^{-1}$ to produce droplets. The droplets passed through a diffusion dryer and a neutralizer to produce dry ammonium sulfate seed particles. We injected seeds with a number mode size of 110 nm until they reached a number concentration of $2\times10^4$ cm$^{-3}$, resulting in an initial seed mass concentration of around 40 µg m$^{-3}$ and a surface area concentration of up to 1000 µm$^2$ cm$^{-3}$. Typical organic vapors with a molar weight of 250 g mol$^{-1}$ thus had an initial collision frequency with these seeds of 0.01 s$^{-1}$. We injected α-pinene (Sigma-Aldrich, ≥ 99 %) into the chamber using a septum injector with purified air as carrier flow. We generated ozone using a corona-discharge ozone generator (AZCO, HTU500AC) to initiate the ozonolysis reaction. We prepared a fresh HONO solution in a bubbler by adding a 4.9 g L$^{-1}$ sulfuric acid solution to a 6.9 g L$^{-1}$ sodium nitrite solution. We then turned on the UV lights to start the photo-dissociation of HONO, producing OH.

At the end of each experiment, we injected additional ammonium-sulfate seeds into the chamber using the same method with a more concentrated solution (5 g L$^{-1}$) in order to characterize the particle wall-loss rates a second time.

We added butanol-d9 (Cambridge Isotope Laboratories, 98 %) into the chamber through the septum injector as an OH tracer before the reaction started and used the method described in Barmet et al. (2012) to calculate the OH produced by HONO photolysis. The OH concentration in these experiments was around $2.4\times10^7$ molecules cm$^{-3}$ for the first hour, then dropped to around $5\times10^6$ molecules cm$^{-3}$ afterwards.



We performed experiments at both low RH of less than 20 % and intermediate RH of 50 %.
To add water vapor to the chamber, we used a stream of purified air to carry ultrapure water
(Millipore water purification system) in a bubbler into the chamber before the introduction of seeds.
We measured the particle size distribution using a TSI Scanning Mobility Particle Sizer,
SMPS (classifier model 3080; CPC model 3010 or 3772), and the particle composition and mass
spectrum of the OA with an Aerodyne High Resolution Time-of-flight Aerosol Mass Spectrometer
(HR-Tof-AMS). We monitored the concentrations of α-pinene and butanol-d9 using a Proton
Transfer Reaction-Mass Spectrometer (PTR-MS, Ionicon), the ozone concentration using a Dasibi
1008 ozone monitor (ICE: Teledyne 400E), and $NO_x$ (NO + $NO_2$) levels using a Teledyne API
$NO_x$ Analyzer 200A (ICE: Teledyne T201). We held the chamber temperature constant at 22 ºC
throughout all experiments. We list the initial conditions of the experiments performed for this
work in Table 1.

**3. Data analysis**
**3.1 SOA yields**
The SOA mass yield, $Y$, is a metric of the ability of a gaseous precursor to form SOA, and
is defined as $Y = C_{SOA}/\Delta VOC$, where $C_{SOA}$ is the produced SOA mass concentration (in µg m$^{-3}$)
and $\Delta VOC$ the amount of the VOC precursor (α-pinene in this case) reacted (in µg m$^{-3}$). To
separate the effect of aging on SOA mass concentration, we define a first-generation SOA mass
yield, $Y_1 = C_{SOA,1}/\Delta VOC$, and a second-generation SOA mass yield, $Y_2 = C_{SOA,2}/\Delta VOC$. $C_{SOA,1}$ and
$C_{SOA,2}$ are the concentrations of SOA formed before, and after aging with hydroxyl radicals. All
α-pinene reacts away during the first stage and thus $\Delta VOC$ for the second stage is the same as the
initial α-pinene concentration in the chamber.

**3.2 Particle wall-loss correction**
In this work, we try to reduce the uncertainties in the estimated SOA mass concentration
associated with the particle wall-loss correction. This uncertainty can be significant due to two
aspects of these aging experiments: the evolution of the particle size distribution and the duration
of the experiments. In these aging experiments, where particles grow by condensation and
coagulation for several hours, the particle size distribution can potentially shift, covering a wide
size range over the course of an experiment. Particle wall losses are size dependent, and this shift



can introduce significant errors if a constant loss rate constant is assumed. To minimize these
problems, we adopted a size-dependent particle wall-loss correction method where we determined
the particle wall-loss rate constant, $k$, at each particle size, $D_p$.

**3.2.1 Determination of particle wall-loss rate constants**

The size-dependent particle wall-loss correction method (Keywood et al., 2004; Ng et al.,
2007; Loza et al., 2012; Nah et al., 2016) adopted in this work is based on the SMPS-measured
particle size distribution. At each particle size bin $i$, the first-order particle wall-loss rate constant
$k$, can be determined as the slope of the following equation:

$$\ln[N_i(t)] = -k_i t + Q \qquad (1)$$


where $N_i(t)$ is the SMPS-measured aerosol number concentration at size bin $i$ and $Q$ is an arbitrary
constant. Applying Eqn. 1 across the entire SMPS-measured particle size range, we obtain the
particle wall-loss rate constant function, $k(D_p)$.
To determine the $k(D_p)$ profile, we utilized the initial four-hour ammonium sulfate seed
wall-loss period for each experiment. Since $k$ may also vary with time (McMurray and Rader,
1985), we determined a second $k(D_p)$ profile for each experiment using the ammonium sulfate
seed wall-loss period at the end. It is important to ensure that the $k$'s, especially at sizes where the
majority of SOA mass is distributed, remain the same over the course of each experiment.
The $k(D_p)$ values calculated (with an $R^2 > 0.5$) based on SMPS measurements of the seed
distribution from this work usually only cover particle size range of 30-300 nm due to the lack of
particles at either end of the particle size distribution. To determine the $k(D_p)$ for $D_p < 30$ nm, we
use a simple log-linear fit of $k$'s from 30-50 nm and back extrapolate it to 10 nm. To determine
$k(D_p)$ for $D_p > 300$ nm, we assume that the constant is practically the same in the 300-600 nm
range. We confirmed this with additional seed-only experiments where there were enough particles
at that size range (Wang et al., 2017). We then applied the complete $k(D_p)$ profile to correct for
the particle number and mass concentration. Details regarding the wall-loss profiles in the CMU
chamber and the execution of the size-dependent particle wall-loss correction for this work can be
found in Wang et al. (2017).



### 3.2.2 Correction of SMPS measurements


The corrected particle number concentration at each size bin $i$, $N_i(t)$, can be calculated
numerically,

$$N_i(t) = \mathrm{N}_i^m(t) + k_i \int_0^t \mathrm{N}_i^m(t)dt, \qquad (2)$$

from the measured values $\mathrm{N}_i^m(t)$ and the $k(D_p)$ corresponding to the size bin $i$, $k_i$.
For closed systems in which coagulation is slow, the particle wall-loss corrected number
concentration should be constant. In order to evaluate how well the correction works, we define
the parameter: $\varepsilon_N = 2\sigma_{N_s}/\overline{N_s}$, where $\sigma_{N_s}$ is the standard deviation of the particle wall-loss
corrected number concentration for the seed wall-loss periods and $\overline{N_s}$ the average. Similarly, we
define $\varepsilon_V = 2\sigma_{N_s}/\overline{V_s}$ based on the particle wall-loss corrected volume concentration for the two
seed wall-loss periods. Only when all four values, $\varepsilon_N$ and $\varepsilon_V$ for both the initial and the final seed
periods, are less than 5 % do we deem the particle wall-loss correction valid for that individual
experiment. Experiments in which these criteria were not met were not included in the analysis.
To calculate the mass concentration of the formed SOA, $C_{SOA}$, during the course of an
experiment, we treated the particle wall-loss corrected aerosol volume concentration $V(t)$
differently before and after its maximum, $V_{max}$ . For

$$\mathrm{t} < t_{V_{max}}, C_{SOA}(t) = (V(t) - V_s)\rho_{SOA},$$
$$\mathrm{t} \geq t_{V_{max}}, C_{SOA}(t) = [V(t) - V_s\frac{V(t)}{V_{max}}]\rho_{SOA}, \qquad (3)$$

where $t_{V_{max}}$ is the corresponding time at the maximum particle wall-loss corrected total aerosol
volume concentration. $V_s$ is the average particle wall-loss corrected seed volume concentration
before the beginning of each experiment. $\rho_{SOA}$ is the SOA density, assumed to be equal to 1.4 µg
m$^{-3}$ (Kostenidou et al., 2007). Ideally, $V$(t) should equal to $V_{max}$ after the reactions are completed
and particle wall loss is the only process after $t_{V_{max}}$. However, deviations of $V$(t) from $V_{max}$ are
caused by the uncertainty associated in applying the size-dependent wall-loss corrections. By
scaling $V_s$ with $V(t)/V_{max}$, we are distributing the impact of any potential fluctuations in $V(t)$ evenly
to both the seeds and the organics, and thus obtain a more stable $C_{SOA}$ after aging.





### 3.3 Analysis of AMS measurements

The HR-AMS was operated in V mode during the experiments in this work. Squirrel v1.56D was used to analyze the data. The atomic oxygen to carbon ratio, O:C, was determined based on the unit-resolution correlation described in Caragaratna et al. (2015). Nitrate signals were attributed to organics since the only sources of them in these experiments are organonitrate compounds.

### 4. Results and discussion

The particle wall-loss corrected aerosol number concentration evolution during a typical experiment (Exp. 1) together with the SMPS raw measurements are shown in Fig. 1. Prior to the ozonolysis, 18,000 cm$^{-3}$ ammonium sulfate particles were added to the chamber as seeds. After a 4.5 h wall-loss period, 8,000 cm$^{-3}$ particles remained suspended, serving as pre-existing surface for condensation. At $t$=0, ozone was added into the chamber, reacting with α-pinene to form condensable first-generation products. An additional 100 cm$^{-3}$ particles were formed due to nucleation at this time. Two doses of HONO were added into the chamber in this experiment at $t$=0.4 h and $t$=1.3 h, respectively. HONO was allowed to mix in the chamber and then the UV lights were turned on at $t$=0.8 h and $t$=1.8 h to produce OH. At $t$=3.5 h, another 10,000 cm$^{-3}$ ammonium sulfate particles were added into the chamber for a second 4 h long determination of the $k(D_p)$ profile for this experiment.

The two $k(D_p)$ profiles determined from the initial seed wall-loss period and the one at the end of the experiment are shown in Fig. 2. They agree relatively well with small discrepancies at $D_p$ < 50 nm. The complete $k(D_p)$ profile used for the size-dependent particle wall-loss correction is also shown.

As indicated in Fig. 1, the particle wall-loss corrected aerosol number concentration remains relative level at t<0 h and t>3.5 h, with $\varepsilon_{N,1}$ = 3.3 % and $\varepsilon_{N,2}$ = 0.5 %, respectively. The particle wall-loss corrected aerosol volume concentration (Fig. 3) at the initial seed wall-loss period and that at the end had variabilities equal to $\varepsilon_{V,initial}$ = 4.2 % and $\varepsilon_{V,end}$ = 3.8 %, respectively. All parameters were less than 5 % and therefore the accuracy of the wall-loss correction was acceptable.

The particle wall-loss corrected aerosol volume concentration evolution for Exp. 1 together with the corresponding SMPS raw measurements are shown in Fig. 3. Particles grew from $t$=0 to



0.7 h and $t$=0.8 to 1 h due to vapor condensation. The total aerosol volume peaked at $t$=0.7 h during
the first-generation oxidation, and reached its maximum at $t$=1.1 h due to aging during the second-
generation oxidation. The change in volume during the second addition of OH at 1.7 h was
negligible.
The SOA mass concentration evolution for Exp. 1 calculated using Eqn. 3 is shown in Fig.
4. The error bars are calculated using the highest ε (in this case $\varepsilon_{V,1} = 4.2$ %). For this experiment,
37.7±1.6 µg m$^{-3}$ of SOA was formed during ozonolysis. An additional 11.1±2.6 µg m$^{-3}$ SOA was
formed during the first aging period. The SOA reached 48.8±2 µg m$^{-3}$ after aging and remained
approximately constant utill the end of the experiment. The total SOA produced and the calculated
SOA yields for all experiments are listed in Table 2.
The AMS-derived atomic oxygen to carbon ratio (O:C) evolution for Exp. 1 is shown
together with the AMS-measured aerosol composition (assuming CE=1) in Fig. 5. The increase in
the sulfate signals at t=0 is caused by a change in the instrument collection efficiency. Due to the
uncertainty caused by CE changes over the course of an experiment, we did not use the absolute
AMS-measured organic mass concentration for any quantitative analysis. Using the algorithm
derived by Kostenidou et al. (2007), we calculated the CE to be ~0.25 for the initial seed period
and ~0.4 after the seeds were coated with organics. A quick check comparing the two stepwise
increase in the CE-corrected organic mass concentration to those derived from SMPS revealed that
the results from both instrument agreed reasonably well.
The O:C is a collective measure for the ongoing chemistry during these aging experiments.
In Exp. 1, the O:C kept decreasing due to the freshly-formed semi-volatile SOA condensing onto
particles from $t$=0 to 0.5 h. From $t$=0.5 h to 0.8 h (UV on), the O:C ratio kept decreasing to 0.42
while the organic mass concentration stayed almost constant. This is consistent with the "ripening"
phenomenon, first observed during the MUCHACHAS campaign, where the composition of the
formed SOA keeps evolving after α-pinene has reacted while the change in SOA mass is minimal
(Tritscher et al., 2011). After OH radicals were generated in the chamber at $t$=0.8 h, the semi-
volatile vapors got oxidized to form second-generation products of lower volatility, resulting in an
increase of 0.02 in O:C in about 10 min. After $t$=1 h, the O:C remained relatively constant but it
started to decrease at $t$=1.25 h when the UV lights were turned off. Since aging is a complex
process that involves functionalization, fragmentation and heterogeneous reactions, the trends in
O:C are indicative of the competition among these processes. The decrease we observed here was





associated with turning the UV lights off, and thus it is likely that some chemistry was perturbed
and thus the processes resulting in decreasing O:C took over. The decrease in O:C associated with
turning off the UV lights was not consistent across the five experiments. This further proves that
this phenomenon is the result of several competing process and needs further investigation on a
molecular level. An inflection point at $t$=1.7 h was observed after a second dose of OH being
introduced in the chamber. Instead of the stepwise increase like the one observed after the first
dose of OH, the O:C increased slowly but steadily this time until the end of the experiment to 0.45
with no significant increase in organic mass. This is also quite consistent with what was observed
in MUCHACHAS.

We used the organic to sulfate ratio (Org/Sulf) derived from AMS measurements to look

at the SOA formation in these experiments due to its insensitivity to changes in collection
efficiency. The Org/Sulf time series for Exp. 1 is shown in Fig. 6. The ratio increased to 1.25 at
t=0.7 h as the result of the first-generation vapors condensing onto pre-existing particles. After we
first turned on the UV lights, a stepwise increase in the ratio was observed and reached the
maximum value of 1.60 at t=1.1 h as a result of the second-generation oxidation chemistry. After
that, the ratio kept decreasing. A small bump was observed after the second introduction of OH
and then the ratio kept decreasing. One possible explanation for this continuous decrease is the
effect of the size-dependent particle wall-loss process. The faster removal of smaller particles
(which contain more SOA than sulfate) than that of the bigger ones (which have a lower SOA to
sulfate ratio) can lead to a decrease of the overall organic to sulfate ratio. Fig. 7 shows the size
dependence of the Org/Sulf, together with the mass distribution of both organic and sulfate for
Exp. 1. The Org/Sulf decreased dramatically from 10 to 1 over the particle vacuum aerodynamic
diameter ($D_{va}$) range of 200 – 500 nm, indicating strong composition dependence on particle size.
Since the majority of the mass is distributed in this range, the size-dependent particle wall-loss rate
can contribute significantly to the decrease observed in Fig. 6 after the Org/Sulf reached its
maximum.

### 4.1 Effect of size-dependent losses on the organic to sulfate ratio

To quantify the effect of the size-dependence of the particle wall-loss process on the

organic to sulfate ratio, we discretized the AMS-measured mass distribution $M(D_p)$ into 10 bins



in the particle diameter space and defined a mass-weighted particle wall-loss rate constant for each
species $j$, $\bar{k}_j$, as

$$\bar{k}_j = \sum_{i=1}^{10} M_{ij} k_i \ / \sum_{i=1}^{10} M_{ij} \qquad (4)$$

where $M_{ij}$ is the aerosol mass concentration of species $j$ for size bin $i$ and $k_i$ is the averaged $k(D_p)$
across size bin $i$. Note that the particle diameter used in this section refers to the SMPS-measured
mobility equivalent diameter $D_p$. The particle vacuum aerodynamic diameters derived from the
AMS measurements have been converted to $D_p$ using an SOA density of 1.4 µg m$^{-3}$.

From Eqn. 4 we are able to determine a mass-weighted particle wall-loss rate constant for

sulfate, $\bar{k}_{SO_4}$, and for organics, $\bar{k}_{Org}$. For the period after completion of the reactions and if there
are only particle losses to the walls the Org/Sulf ratio should satisfy:

$$(\text{Org/Sulf})(t) = (\text{Org/Sulf})_m(t) \exp\left(\bar{k}_{SO_4} - \bar{k}_{Org}\right)t \qquad (5)$$

where $(\text{Org/Sulf})_m(t)$ is the AMS-measured and $(\text{Org/Sulf})(t)$ the loss-corrected organic to
sulfate ratio.

We can test if indeed the particle wall losses are responsible for the decreasing ratio in Exp.

1 focusing on the period from $t_1 = 1.2\,\text{h}$ to $t_2 = 1.7\,\text{h}$ (Fig. 6). In this example $t_1$ corresponds to
the maximum Org/Sulf and $t_2$ is the second time in which the UV lights were turned on. Applying
Eqn. 4, we found the mass-weighted particle wall-loss rate constant for organics, $\bar{k}_{Org} = 0.06\,h^{-1}$,
and for sulfate, $\bar{k}_{SO_4} = 0.05\,h^{-1}$. The black line in the inset graph of Fig. 6 indicates the particle
wall-loss corrected Org/Sulf for the chosen time period using Eqn. 5. The loss-corrected ratio
remained relatively constant indicating that the size-dependent particle wall-loss process coupled
with the different size distributions of the sulfate and organics were causing the decrease in the
ratio. This exercise was repeated for the other experiments arriving in the same conclusion.

**4.2 Effect of chemical aging on additional SOA formation**

To quantify aging effects based on the SMPS measurements, we define the fractional

change in the particle wall-loss corrected SOA mass concentration after aging, Δ[OA] , as:




$$\Delta[OA] = (C_{SOA,2} - C_{SOA,UV})/C_{SOA,1}, \tag{6}$$


where $C_{SOA,UV}$ is the particle wall-loss corrected aerosol mass concentration at the time when we
first turned on the UV lights. $C_{SOA,\ UV}$ can be equal to $C_{SOA,1}$ depending on how level the first-
generation SOA mass concentration remains after wall-loss correction. Fig. 8 summarizes the
$\Delta[OA]$ for all five experiments with the values and corresponding errors listed in Table 2. The OH
exposure resulted in an average increase of 24±6 % in SOA mass concentration after aging, ranging
from 20 to 29 %. Our HONO injection method creates OH levels of about $2.4 \times 10^7$ molecules
cm$^{-3}$ for the first hour and then the concentration dropped to around $5 \times 10^6$ molecules cm$^{-3}$. The
OH exposure is equivalent to 2-4 days of typical atmospheric oxidation conditions, assuming an
OH concentration of $2 \times 10^6$ molecules cm$^{-3}$. The uncertainties displayed in Fig. 8 were propagated
from uncertainties in the SOA mass concentration.
To quantify aging effects based on the AMS data, we define the fractional change in the
organic to sulfate ratio:

$$\Delta[Org/Sulf] = ([Org/Sulf]_2 - [Org/Sulf]_{UV})/[Org/Sulf]_1, \tag{7}$$


where $[Org/Sulf]_{UV}$ refers to the organic to sulfate ratio at the time when we first turned on the UV
lights, $[Org/Sulf]_1$ the maximum before we first turned on the UV lights and $[Org/Sulf]_2$ the
maximum after the OH exposure. Fig. 8 summarizes the $\Delta[Org/Sulf]$ calculated for all five
experiments with the values and corresponding errors listed in Table 2. The uncertainties are based
on the deviation between the measured and the corrected Org/Sulf (Fig. 6 inset) over the chosen
time period. An associated error is calculated respectively for $[Org/Sulf]_{UV}$, $[Org/Sulf]_1$ and
$[Org/Sulf]_2$. The reported error for $\Delta[Org/Sulf]$ in Table 2 is the propagated results of the three.
For experiments in this work, the percent increase in organic to sulfate ratios ranged from 18 to
27 % with an average increase of 21±4 %. The values are fairly consistent with the SMPS-derived
$\Delta[OA]$.

**4.2.1 Role of RH**
Exp. 5, performed at the intermediate RH of 50 %, resulted in a comparable change in SOA
formation after aging as experiments at lower RH (Fig. 8). In this experiment, the increase in





Org/Sulf after aging was 21.2 %, 1.5 % higher than the average Δ[Org/Sulf] of experiments 2-4.
Δ[OA] for Exp. 5 was 20.5 %, about 2 % lower than the average Δ[OA] of experiments 2-4. The
effect of RH on the SOA formation during chemical aging, at least for these conditions, appears
to be small.

**4.2.2 Role of organic vapor loss to the Teflon walls**

For chamber SOA experiments with preexisting particles, the particles act as competing

surface against the chamber walls. We calculated the condensation sink (CS) of particles using the
method described in Trump et al. (2014) with a unit accommodation coefficient, consistent with
recent findings (Julin et al., 2014; Palm et al., 2016). The calculated condensation sink in the form
of time scale for vapors condensing onto particles (1/CS) for Exp. 1 is shown in Fig. 9. During the
entire experiment, the timescale for vapors to condense onto particles remained less than a minute.
Compared to the organic vapor wall-loss timescale of 15 min in the CMU chamber (Ye et al.,
2016), the vapors condense onto the particles 15 times faster than that onto the walls. This
corresponds to a 6.3 % loss of the semi-volatile vapors to the walls. For the experiments conducted
in this work, the yields should be increased by 1-3 % after accounting for the vapor wall-loss effect.

The situation is a little more complex for the second-generation oxidation because material

with higher volatilities that could have become SOA were lost during the time after the end of the
first phase and before the beginning of the second. To address this issue, OH radicals were
introduced about an hour earlier in Exp. 1 as compared to the rest of the experiments. A shorter
timescale ensures the first-generation vapor products react efficiently with OH instead of
interacting with the chamber walls as in the case of longer timescales. There was an increase of
27 % in Org/Sulf in this experiment after aging, 7 % more than the average of the other four
experiments. Δ[OA] for Exp. 1 was 29.4 %, about 7.5 % higher than the average of the rest four
experiments. If we attribute this 7 % difference purely to the vapor wall-loss effect, then we
estimate that vapor losses can increase the additional SOA formation by roughly another 10 % for
the experiments conducted in this work.

**4.3 Effect of chemical aging on O:C**

Fig. 10 summarizes the absolute increase in O:C after the two doses of OH, respectively,

with the corresponding exposure required to achieve the increase. As we discussed above using



Exp. 1 as an example, the O:C in all experiments showed a stepwise increase after the first OH
introduction while it grew continuously after the second OH introduction until the end of the
experiment. For these five experiments, it took 10 - 30 min for the O:C to increase by 0.02-0.04.
The stepwise increase in O:C is caused by the rapid reactions between the first generation vapor
products and the OH. One of the major products identified in the gas phase from the α-pinene
ozonolysis system, pinonaldehyde, reacts with OH at a rate of $4.4 \times 10^{-11}$ cm$^3$ molecule$^{-1}$ s$^{-1}$
(Atkinson and Arey, 2003). During the first hour of OH introduction, the OH concentration
remains on average at a steady state of $2.4 \times 10^7$ molecule cm$^{-3}$. A quick estimation of $1/k_{OH}[OH]$
gives a timescale of 16 min, which is consistent with what we observed in these experiments. The
second exposure corresponds to the period until the end of each experiment. The increase in O:C
of 0.01 to 0.04 clearly indicates change in SOA composition, however paired with minimum
change in SOA mass. This phenomenon is likely caused by heterogeneous reactions.

**4.4 Comparison with other studies**
Overall, the results from our chamber experiments in this work are consistent to those from
the MUCHACHAS chambers. After adopting a size-dependent particle wall-loss correction
method, we observed 20-30 % additional SOA formation after aging. Vapor wall-loss effect can
account for an additional 10 %, increasing the range to 20-40 %. The O:C presented a stepwise
increase of 0.02-0.04 after the first introduction of OH, and then increased gradually overtime after
the second introduction of OH.
During the MUCHACHAS campaign, mixtures of SOA and gas-phase products formed in
the Paul Scherrer Institute (PSI) 27 m$^3$ Teflon chamber from low (10 ppb) and high (40 ppb) initial
α-pinene concentration were exposed to OH by TME ozonolysis and HONO photolysis at an RH
of approximately 50 % (Tritscher et al. 2011). An OH concentration of $2 \times 10^6$ to $10 \times 10^6$ molecules
cm$^{-3}$ was maintained up to four hours. The authors reported an additional 50 % SOA mass forming
after aging using the first-order, size-independent particle wall-loss correction for the suspended
organic mass concentration measured by AMS. An increase of 0.04 in the oxygen to carbon ratio
was also observed during aging.
In the 84.5 m$^3$ Aerosol Interaction and Dynamics in the Atmosphere (AIDA) aluminum
chamber at Karlsruhe Institute of Technology, an OH concentration of $2 \times 10^6$ to $10 \times 10^6$ molecules
cm$^{-3}$ was used by a constant flow of TME (dark aging). The authors observed an increase of 17-



55 % in the SMPS-derived SOA mass concentration (density corrected) after aging during four experiments with initial α-pinene concentration ranging from 14 to 56 ppb (Salo et al., 2011). In the 270 m$^3$ Simulation of Atmospheric Photochemistry in a large Reaction (SAPHIR) Teflon chamber at Forchungzentrum Jülich, SOA and vapors generated from the ozonolysis of 40 ppb α-pinene was aged for three consecutive days with OH produced by ambient light chemistry. An OH concentration of 2-5×10$^6$ molecules cm$^{-3}$ was maintained and 9 %, 4 % and 1 % additional SOA was formed respectively after aging each day. These values were corrected for particle wall loss using different wall-loss rate constants determined during different periods of the experiment.

Our result of 20-40 % additional SOA formation due to aging is well within the range of that from the above chambers. The difference in the results from each chamber could potentially be attributed to different OH exposure (e.g. a constant flow of HONO or TME was provided in the PSI chamber). Other plausible explanations include whether the reported values were particle wall-loss corrected and whether the same method was adopted for the correction.

For the HONO aging experiment performed in the CMU chamber during the MUCHACHAS campaign, Henry and Donahue (2012) suggested a potentially strong photolysis effect based on decreasing organic to sulfate ratio derived from the AMS measurements. In our experiments, the organic to sulfate ratio was affected by the size-dependent wall-loss process. Both the AMS-measured organic to sulfate ratio and the SMPS-measured OA remained relatively constant after correcting for the size dependence of the particle-wall process in these experiments. We thus conclude that minimum photolysis was observed for our experiments.

## 5. Conclusions

With an OH exposure equivalent to 2-4 days of typical atmospheric oxidation conditions, the OH aging of the α-pinene ozonolysis products formed 20-40 % additional SOA mass for the experimental conditions used in this work. Elevated RH up to 50 % has minimum effect on SOA production due to aging. We have constrained the aging effects on additional SOA formation quantitatively using both SMPS and AMS measurements.

A more oxygenated product distribution was observed after aging. A stepwise increase of 0.02-0.04 in O:C was observed within half an hour after the first introduction of OH. After the second-generation products were exposed to additional OH, the O:C grew continuously until the end of the experiments with an absolute increase of up to 0.04. During this period, minimum SOA





production was observed. We attribute this phenomenon to condensed-phase reactions. Further
investigation on a molecular scale is needed.

*Acknowledgement:* The work was funded by the EPA STAR grant 835405 and the EUROCHAMP-
2020 EU project.

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




**Table 1:** Initial conditions of the α-pinene ozonolysis aging experiments.


| Experiment | α-pinene (ppb) | O$_3$ (ppb) | Initial seed surface area ($\mu m^2$ cm$^{-3}$) | RH (%) | OH[a] ($\times 10^7$ molecules cm$^{-3}$) | OH introduction time (h after α-pinene consumption) |
|---|---|---|---|---|---|---|
| 1 | 33 | 450 | 850 | <20 | 2.4 | 0.3 |
| 2 | 14 | 600 | 760 | <20 | 2.7 | 0.8 |
| 3 | 35 | 450 | 720 | <20 | 2.0 | 1.1 |
| 4 | 16 | 500 | 950 | <20 | 2.4[b] | 1.1 |
| 5 | 20 | 400 | 710 | ~50 | 2.7 | 0.8 |


[a]The OH concentration was calculated using the decay of butanol-d9 (monitored by PTRMS) (Barmet et al., 2012).

[b]Estimated OH concentration for Exp. 4 based on the other experiments. The PTRMS data was not available during that time for Exp. 4.




















**Table 2:** SOA mass concentration and yields of the α-pinene ozonolysis aging experiments.

| Experiment | $C_{SOA,1}$ (µg m$^{-3}$) | $Y_1$ (%) | $C_{SOA,2}$ (µg m$^{-3}$) | $Y_2$ (%) | ΔOA (%) | Δ[Org/Sulf] (%) |
|---|---|---|---|---|---|---|
| 1 | 37.7±1.6 | 20.6±0.9 | 48.8±2.0 | 26.7±1.1 | 29.4±6.9 | 27.0±5.8 |
| 2 | 16.7±0.9 | 21.5±1.2 | 18.3±1.0 | 23.5±1.3 | 19.8±8.1 | 18.1±2.9 |
| 3 | 57.1±1.3 | 29.4±0.7 | 71.0±1.6 | 36.2±0.8 | 23.5±3.6 | 19.1±3.6 |
| 4 | 16.8±0.6 | 19.1±0.6 | 20.8±0.7 | 23.7±0.8 | 24.0±5.3 | 21.9±2.1 |
| 5 | 22.2±0.7 | 19.5±0.6 | 25.4±0.8 | 22.3±0.7 | 20.5±4.7 | 21.2±4.4 |






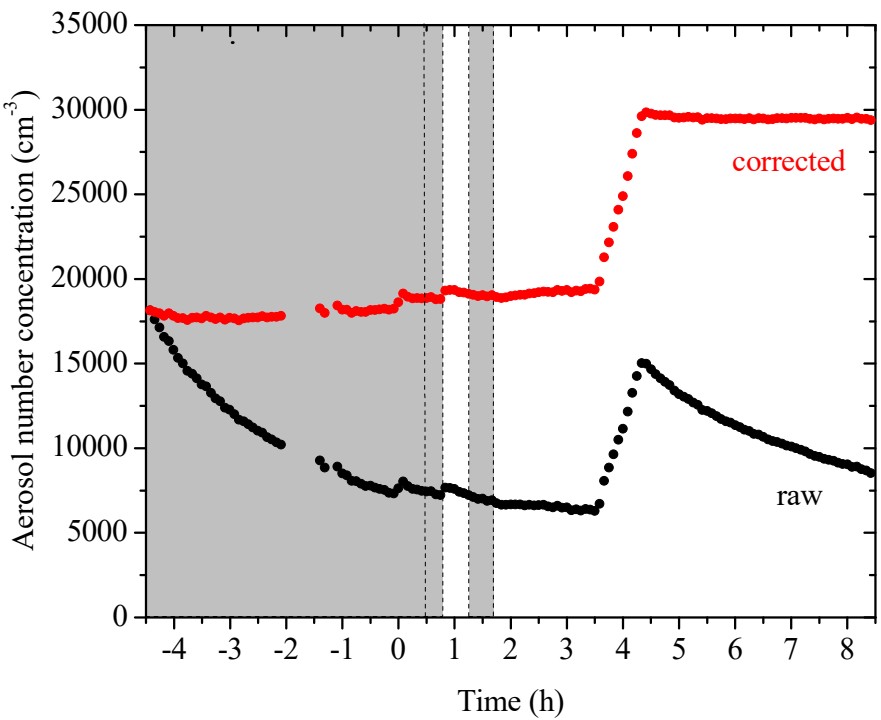


**Figure 1:** SMPS-measured (black symbols) and the size-dependent particle wall-loss corrected
(red symbols) aerosol number concentration evolution during a typical experiment (Exp. 1). Ozone
was added into the chamber at time zero to initiate α-pinene ozonolysis. The shaded areas indicate
that the chamber was dark. The dashed lines mark the beginning and the end of the two times
HONO were added, respectively. The increase in number concentration at t=3.5 h is due to the
injection of 5 g L$^{-1}$ ammonium sulfate particles. An additional 100 cm$^{-3}$ particles were formed due
to nucleation both at the ozonolysis step and the aging step. Data were not recorded from t = -2 h
to -1.4 h.




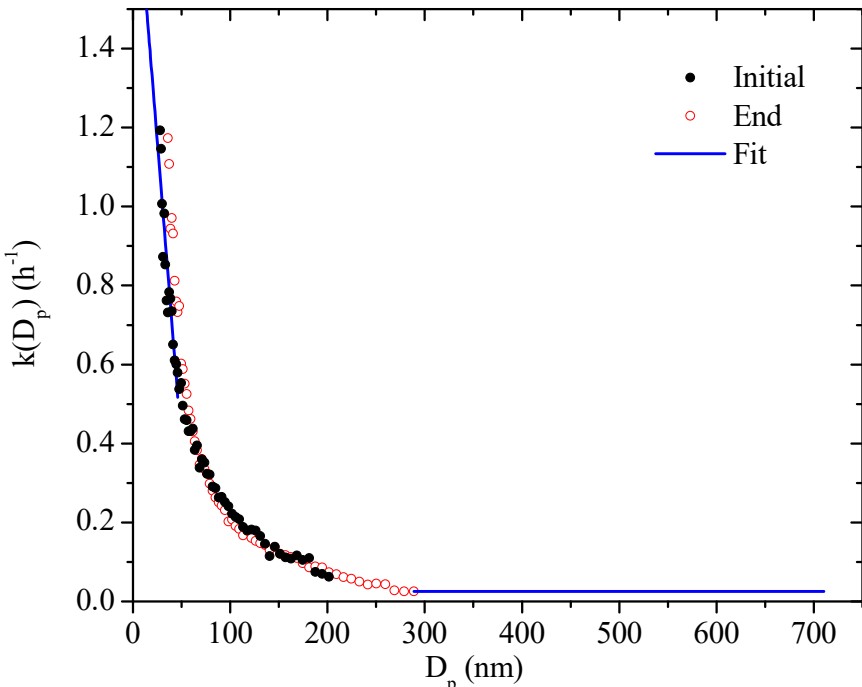

**Figure 2:** The size-dependent particle wall-loss rate constant profile, $k(D_p)$, for Exp. 1. The black
symbols are the rate constants calculated based on the wall-loss process of the initial ammonium
sulfate seed particles from t=-4.5 h to t=0 h, while the red open symbols that of the additional
ammonium sulfate particles at the end from t=4.5 h to t=8.5 h. The blue line is the fit determined.





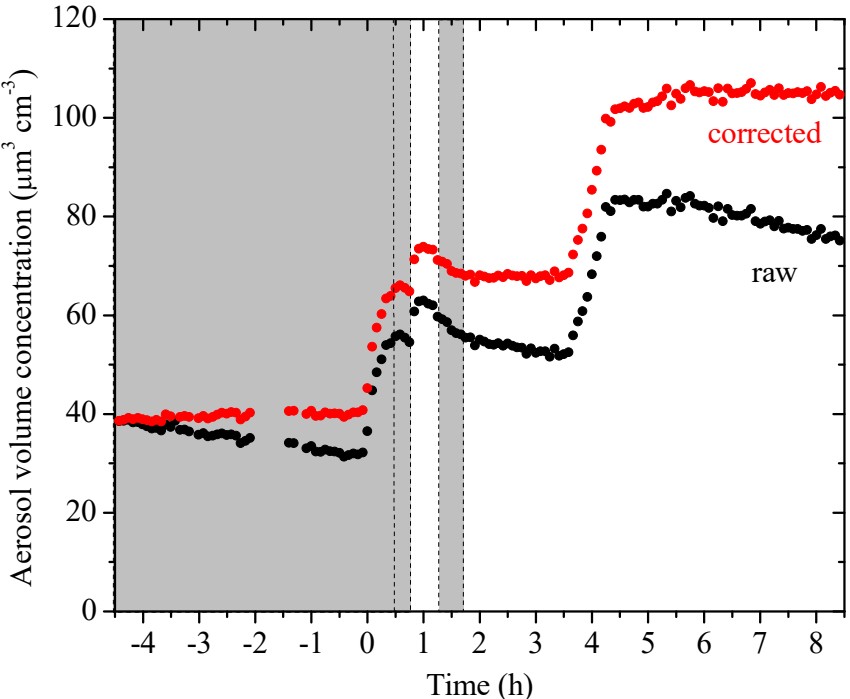


**Figure 3:** SMPS-measured (black symbols) and the size-dependent particle wall-loss corrected (red symbols) aerosol (seed and organic) volume concentration evolution during a typical experiment (Exp. 1). Ozone was added into the chamber at time zero to initiate α-pinene ozonolysis. The shaded areas indicate that the chamber was dark. The dashed lines mark the beginning and the end of the two times HONO were added, respectively. 5 g $L^{-1}$ ammonium sulfate particles were injected into the chamber at t=3.5 h. Data were not recorded from t=-2 h to -1.4 h.







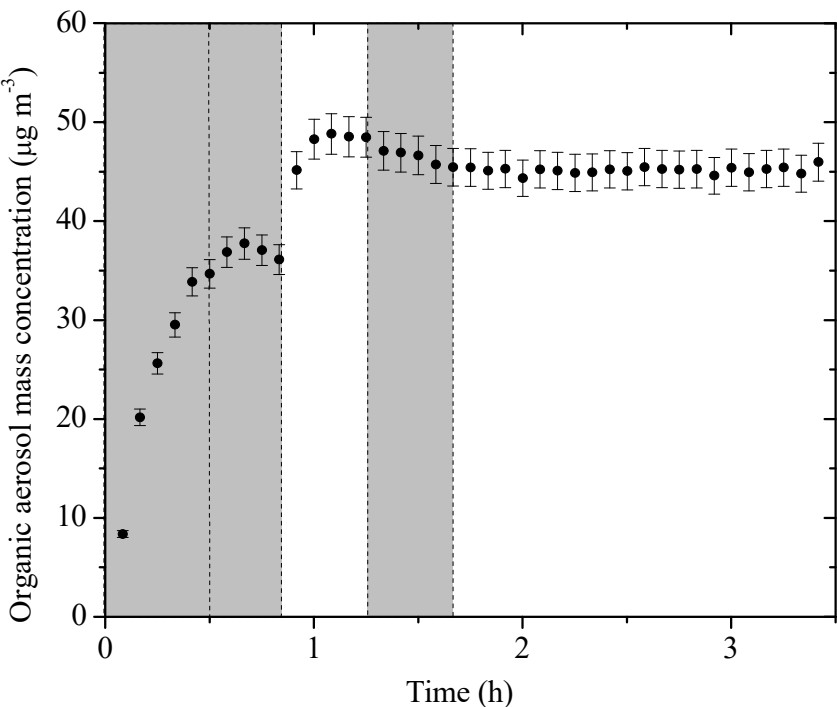

**Figure 4:** The particle wall-loss corrected SOA mass concentration ($\rho$=1.4 g cm$^{-3}$) evolution for
Exp. 1 derived from SMPS measurements. The corresponding error shown is due to the particle
wall-loss correction. Ozone was added into the chamber at time zero to initiate α-pinene ozonolysis.
The shaded areas indicate that the chamber was dark. The dashed lines mark the beginning and the
end of the two times HONO were added, respectively.








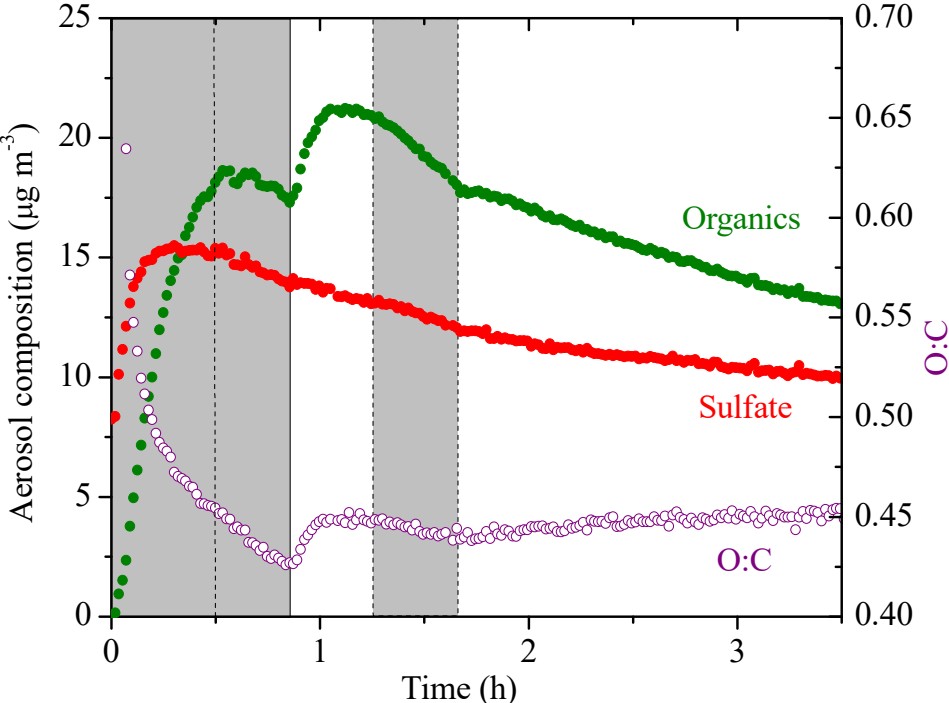

**Figure 5:** The AMS-measured aerosol composition (CE=1) (left axis) and the atomic oxygen to carbon ratio (right axis) evolving with time for Exp. 4. The increase in the sulfate signal at t=0 is the result of a change in the collection efficiency (CE). Ozone was added into the chamber at time zero to initiate α-pinene ozonolysis. The shaded areas indicate that the chamber was dark. The dashed lines mark the beginning and the end of the two times HONO were added, respectively.

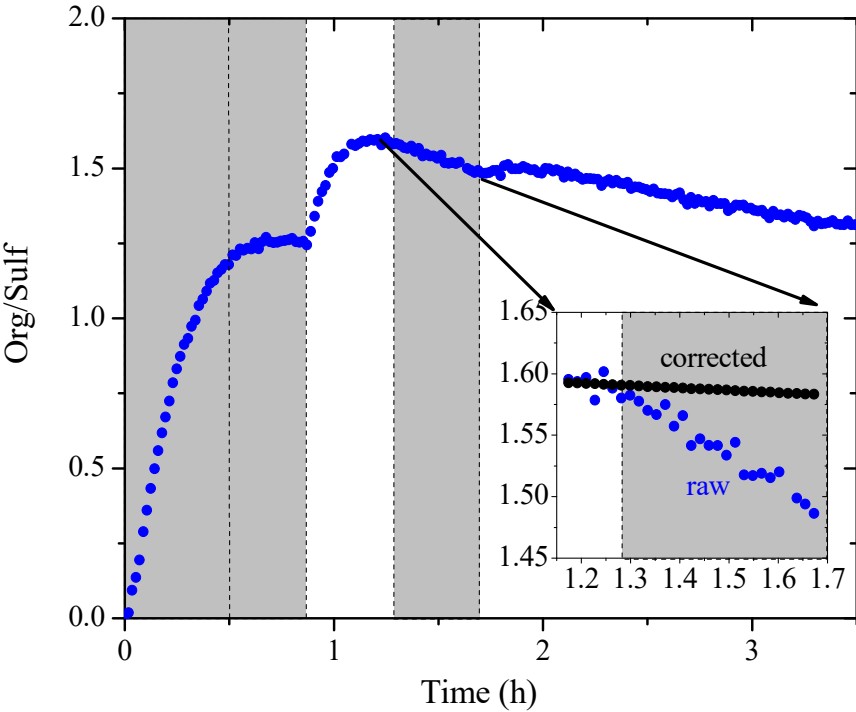

713
**Figure 6:** The AMS-derived organic to sulfate ratio time series for Exp. 1. The inset is a blow-up
of the Org/Sulf ratio from its maximum until the second time when the UV lights were turned on.
The black symbols are the particle wall-loss corrected Org/Sulf during that half hour. Ozone was
added into the chamber at time zero to initiate α-pinene ozonolysis. The shaded areas indicate that
the chamber was dark. The dashed lines mark the beginning and the end of the two times HONO
were added, respectively.




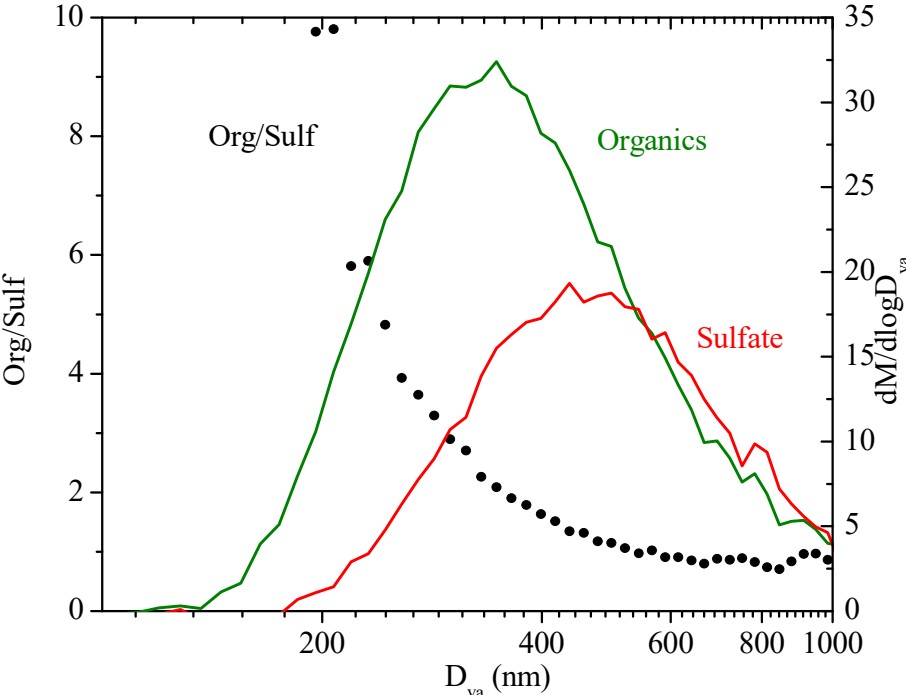

720

**Figure 7:** The dependence of the AMS-derived organic to sulfate ratio on particle vacuum
aerodynamic diameter for Exp. 1 (left axis). Also shown are the AMS-measured organic (green)
and sulfate (red) mass distribution (right axis). The results are based on PToF data averaged over
~2.5 hours (t=1.1 h to 3.5 h).











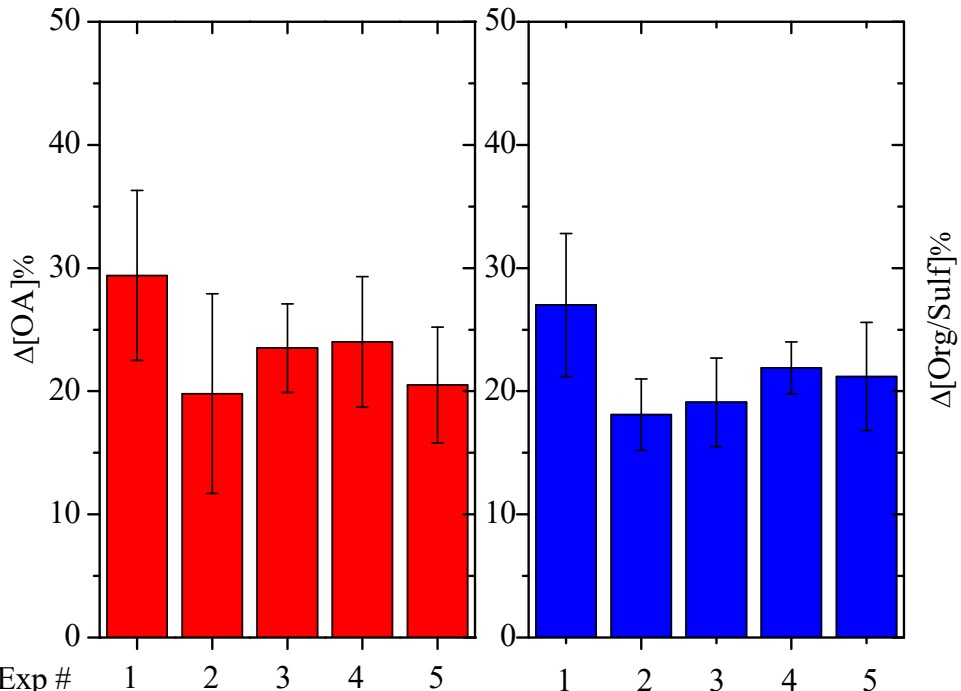



**Figure 8:** SMPS-derived percent change in the particle wall-loss corrected SOA (red columns) mass concentration after aging and AMS-derived percent change in organic to sulfate ratio (blue columns) after aging for all five experiments.











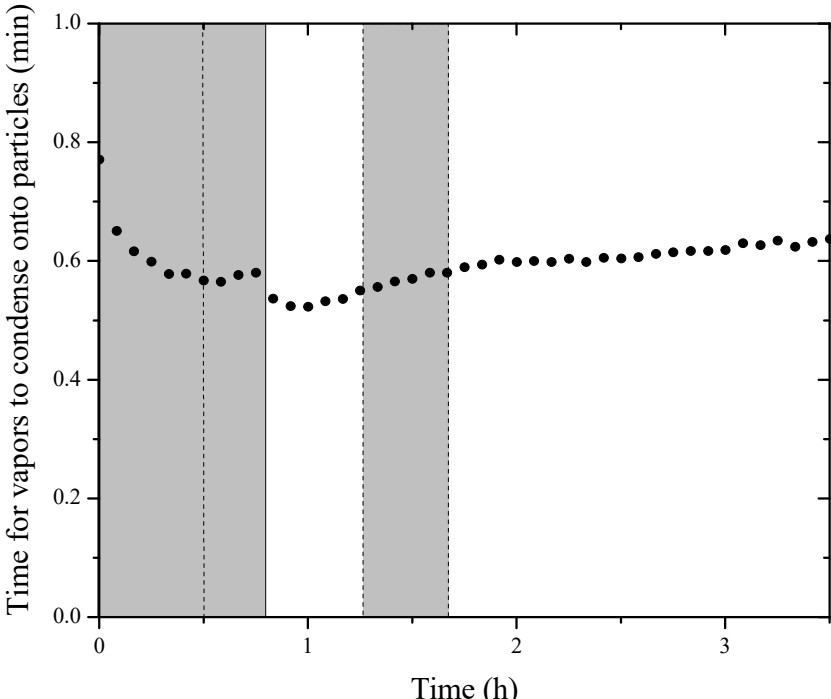


**Figure 9:** The calculated condensation sink (CS) in the form of time scale for vapors condensing onto particles (1/CS). Ozone was added into the chamber at time zero to initiate α-pinene ozonolysis. The shaded areas indicate that the chamber was dark. The dashed lines mark the beginning and the end of the two times HONO were added, respectively.



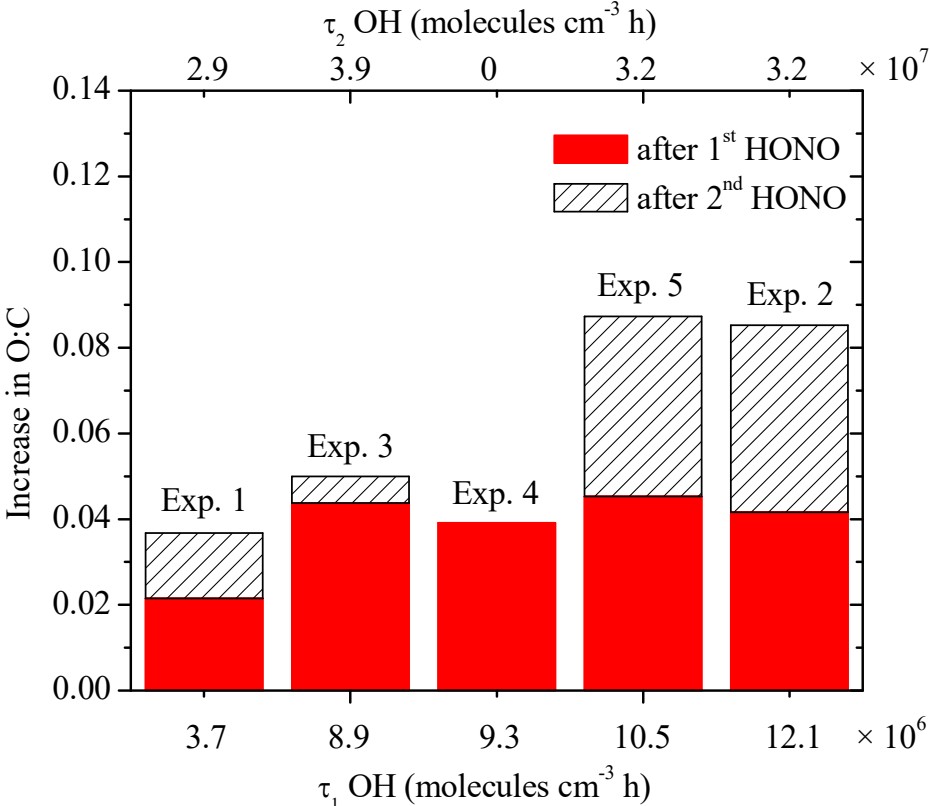

747

748

**Figure 10:** The absolute increase in O:C after the two doses of OH, respectively, with the corresponding exposure. The solid red columns are the increase in O:C after the first introduction of OH, with the corresponding exposure on the bottom axis. The hatched columns are the increase in O:C after the second introduction of OH, with the corresponding exposure on the top axis.






