# Peer review of "Multi-generation Chemical Aging of α-Pinene Ozonolysis Products by Reactions with OH"

_Atmospheric Chemistry and Physics, 2017_

## Referee Comment (RC1) · Anonymous Referee #1 · 20 Sep 2017

This study investigated the OH-oxidation of α-pinene ozonolysis products by reporting the mass yields and elemental composition of SOA produced. The main finding of this study is that OH-oxidation of α-pinene ozonolysis products that is equivalent of 2-4 days of atmospheric OH exposure leads to 20-40% net increase in the SOA yields and an increase in the aerosol O:C ratio by up to 0.04. While highlighting the importance of multi-generation aging in the atmospheric evolution of SOA, a topic that has been extensively studied previously, this finding alone, however, does not warranty the scientific significance and publication of this manuscript on ACP. Substantial revisions are needed, as described in the individual points below.

**General:**

To what extent the experimental observations that 20-40% increase in SOA yields by further OH-oxidation could be applied to the actual atmospheric conditions? The authors used HONO as the OH precursor, which means that hundreds of ppbv levels of NO were present in the experiments conducted. Such an experimental condition is quite different from what is in the atmosphere, where most monoterpene emissions are from remote regions that encounter low NO air masses (e.g., a few ppb or less). As we know, NO significantly alters the VOCs oxidation mechanisms primarily by reacting with $RO_2$ radicals, leading to vastly different product distributions from those observed in the absence of NO, a chemical regime where $RO_2+HO_2$ and $RO_2+RO_2$ reactions dominate the fate of $RO_2$ radicals. That said, the experimental results presented in this study only represent a barely-seen scenario in the atmosphere and does not have any atmospheric relevance. The authors are suggested to conduct 'low-$NO_x$' experiments as well where $H_2O_2$, for example, can be used as the OH precursor. The observed increase in SOA yields, together with the high-NO experiments, can be used as upper and lower limits for the a-pinene SOA aging in the atmosphere.

The methodology of this study, including the chamber experimental approach, SOA yield measurements corrected by particle wall loss, O:C ratio calculations by AMS measurements, has been widely used in the community for decades. The main result of this study, i.e., SOA yields increase by 20%-40% upon OH-oxidation aging, is insufficient to warranty publication of this manuscript. The authors should endeavor to

explore more detailed mechanisms involved in the a-pinene SOA aging by thoroughly analyzing the AMS data, e.g., What functionalities/products have changed and contributed to SOA production during aging? Any indication of PMF analysis, for example, on the key processes involved in the chemical aging?

**Specific:**

Page 2, Line 35-36: There have been a number of studies investigating the SOA aging processes using both static chamber and flow tube reactors (e.g., Robinson et al., Science, 2007; Loza et al., ACP, 2012; Lambe et al., ES&T, 2012). The authors need to revise the statement 'Most laboratory studies of SOA formation so far have focused on the ...'

Page 3, Line 64: References need to be given here.

Page 6, Line 154: What is the particle size range measured by SMPS?

Page 6, Line 171: The OH radical molar yield from ozonolysis of a-pinene is roughly 0.7. So technically, the measured SOA yield at first stage is already a result of certain OH aging. Have the authors estimated the fraction of a-pienne that was oxidized by OH radical in the ozonolysis experiments? Have the authors considered adding any OH scavengers (e.g., $H_2O_2$, CO, methanol) in the first stage of the experiments?

Page 7, Line 205: The assumption that the particle loss rate in the 300-600 nm range is the same potentially leads to underestimation of the corrected SOA yields, as bigger particles have higher wall loss rate due to gravity deposition.

Page 8, Line 236: The authors are suggested calculate the SOA density from the AMS measured O:C and H:C ratio of SOA (Kuwata et al., ES&T, 2012) and compare with the current value used.

Page 9, Line 237-238: Adding the '$V(t)/V_{max}$' factor in the SOA yield correction procedure does not seem necessary. The authors stated that 'deviations of $V(t)$ from $V_{max}$ are caused by the uncertainty associated in applying the size dependent wall loss corrections'. As we know, particle wall loss rate depends on a few parameters, including chamber size, eddy diffusion, static charges on the Teflon surface, and etc. For most experiments presented here that only lasted for a few hours, these parameters should be

fairly constant so that the particle wall loss rate should be quite consistent before and after one experiment. Where are the deviations originating from? I wonder if the assumption that particle loss rate in the 300-600 nm range is the same has any impact on the SOA yield correction here.

Page 27, Figure 5: Why the O:C ratio of SOA kept decreasing in the second dark period? What are the concentrations of $NO_x$ and $O_3$ during this period? Is there any $NO_3$-initiated chemistry going on?

---

## Referee Comment (RC2) · Anonymous Referee #2 · 27 Sep 2017

General Comments

In this manuscript the authors present results of a laboratory study in which they investigated the chemical aging of SOA formed from the ozonolysis of a-pinene. The aging was conducted by photolyzing HONO (to form OH radicals) that was added at two different times following completion of the ozonolysis reaction. The effects of aging were determined by monitoring the SOA mass using an SMPS and the composition using an AMS. Careful corrections were made for the effects of particle and vapor wall loss on the SOA yields. The results demonstrate that oxidation comparable to a few days of atmospheric aging lead to the formation of a significant amount of additional SOA mass, as well as small changes in composition as measured by the O/C ratio. The results are consistent with previous studies of this type, but are of higher quality with regards

to SOA yield corrections that are important in aging experiments. The experiments and data analysis were carefully done, and the manuscript is well written. I think the manuscript will be suitable for publication in ACP once the following comments have been addressed.

Specific Comments

1. Lines 407–410: It seems that this estimate of the effect of gas-wall partitioning of vapors on SOA formation assumes that all the first-generation products are either fully volatile or non-volatile with respect gas-particle partitioning. But for semi-volatile compounds the 15 min condensation time scale is also the upper limit to the time scale to achieve gas-particle partitioning equilibrium. In this case, some vapor will remain in the gas phase and continue to be lost to the chamber walls throughout the experiment because of the large effective particle mass of the walls. I think the approach used here thus provides only a lower-limit estimate to loss of vapors to the wall.

2. Lines 408–410: It is not clear how the estimated loss of vapors to the walls is converted to an SOA yield correction.

3. Line 435–437: It would be straightforward to estimate the potential increase in O/C ratio due to heterogeneous oxidation by OH radicals and thus test this hypothesis.

4. Line 437: Why can't the observed changes in O/C ratio with aging be the result of gas-phase oxidation of semi-volatile compounds coupled to gas-particle partitioning, as proposed by Robinson et al. (2007), rather than heterogeneous oxidation?

5. The Conclusions section as is really just a brief summary of the results, written solely within the context of these experiments. I suggest the authors provide a broader discussion of the relevance of these results to studies of atmospheric SOA and what they contribute to knowledge of the formation, composition, and properties of SOA.

Technical Comments

None.

---

## Author Comment (AC1) · 5 Dec 2017

**(1)** *In this manuscript the authors present results of a laboratory study in which they investigated the chemical aging of SOA formed from the ozonolysis of a-pinene. The aging was conducted by photolyzing HONO (to form OH radicals) that was added at two different times following completion of the ozonolysis reaction. The effects of aging were determined by monitoring the SOA mass using an SMPS and the composition using an AMS. Careful corrections were made for the effects of particle and vapor wall loss on the SOA yields. The results demonstrate that oxidation comparable to a few days of atmospheric aging lead to the formation of a significant amount of additional SOA mass, as well as small changes in composition as measured by the O/C ratio. The results are consistent with previous studies of this type, but are of to SOA yield*

[Figure]

*corrections that are important in aging experiments. The experiments and data analysis were carefully done, and the manuscript is well written. I think the manuscript will be suitable for publication in ACP once the following comments have been addressed.*

We address the various comments of the reviewer below. Our responses and corresponding changes in the paper follow each comment.

*Specific:*

**(2)** *Page 14, Lines 407–410: It seems that this estimate of the effect of gas-wall partitioning of vapors on SOA formation assumes that all the first-generation products are either fully volatile or non-volatile with respect gas-particle partitioning. But for semivolatile compounds the 15 min condensation time scale is also the upper limit to the time scale to achieve gas-particle partitioning equilibrium. In this case, some vapor will remain in the gas phase and continue to be lost to the chamber walls throughout the experiment because of the large effective particle mass of the walls. I think the approach used here thus provides only a lower-limit estimate to loss of vapors to the wall.*

The reviewer is correct. Our approach yields the lower-limit estimate to vapor wall loss, and yet this is consistent with what we observed from the measurements. As indicated in Fig. 6, the organics to sulfate ratio stayed practically constant after its first peak at t=0.7 h until the introduction of OH. This is consistent with the fact that the SVOCs formed in our system only accounted for a small fraction of the products. This is also consistent with what Ye et al. (2015) observed for the $\alpha$-pinene ozonolysis system. With a moderate precursor concentration ($\alpha$-pinene <90 ppb), the SVOCs formed represented 20 percent of the products. We have added this discussion to the paper.

**(3)** *Page 14, Lines 408–410: It is not clear how the estimated loss of vapors to the walls is converted to an SOA yield correction.*

Our zeroth-order correction simply assumes that the yields increase also by 6.3 percent. This results in an absolute increase of the yields of 1-3 percent. We have added this clarification to the paper.

**(4)** *Page 15, Line 435–437: It would be straightforward to estimate the potential increase in O/C ratio due to heterogeneous oxidation by OH radicals and thus test this hypothesis.*

Heterogeneous oxidation here refers to the uptake of OH by the SVOCs and the LVOCs in the condensed phase and the corresponding reactions that take place. Given that these reactions can be quite complex probably involving both functionalization and fragmentation the estimation of the expected O/C ratio is not straightforward.

**(5)** *Page 15, Line 437: Why can't the observed changes in O/C ratio with aging be the result of gas-phase oxidation of semi-volatile compounds coupled to gas-particle partitioning, as proposed by Robinson et al. (2007), rather than heterogeneous oxidation?*

Gas-phase reactions could of course contribute to the observed O/C changes. However, the corresponding condensation of the products should result in a detectable increase in SOA concentration during the same period. We could not observe such a change; therefore, the contribution of gas-phase oxidation is probably small. Heterogeneous reactions can explain this significant change of O/C without a corresponding increase in SOA mass concentration. We added this explanation to the paper.

**(6)** *Page 16: The Conclusions section as is really just a brief summary of the results, written solely within the context of these experiments. I suggest the authors provide a broader discussion of the relevance of these results to studies of atmospheric SOA and what they contribute to knowledge of the formation, composition, and properties of SOA.*

We have followed the suggestion of the reviewer and added a discussion of the relevance of these results for atmospheric SOA.

---

## Author Comment (AC2) · 5 Dec 2017

**(1)** *This study investigated the OH-oxidation of a-pinene ozonolysis products by reporting the mass yields and elemental composition of SOA produced. The main finding of this study is that OH-oxidation of a-pinene ozonolysis products that is equivalent of 2-4 days of atmospheric OH exposure leads to 20-40 percent net increase in the SOA yields and an increase in the aerosol O:C ratio by up to 0.04. While highlighting the importance of multi-generation aging in the atmospheric evolution of SOA, a topic that has been extensively studied previously, this finding alone, however, does not warranty the scientific significance and publication of this manuscript on ACP. Substantial revisions are needed, as described in the individual points below.*

[Figure]

We address the various comments of the reviewer below. Our responses and corresponding changes in the paper follow each comment.

*General:*

**(2)** *To what extent the experimental observations that 20-40 percent increase in SOA yields by further OH-oxidation could be applied to the actual atmospheric conditions? The authors used HONO as the OH precursor, which means that hundreds of ppbv levels of NO were present in the experiments conducted. Such an experimental condition is quite different from what is in the atmosphere, where most monoterpene emissions are from remote regions that encounter low NO air masses (e.g., a few ppb or less). As we know, NO significantly alters the VOCs oxidation mechanisms primarily by reacting with $RO_2$ radicals, leading to vastly different product distributions from those observed in the absence of NO, a chemical regime where $RO_2+HO_2$ and $RO_2+RO_2$ reactions dominate the fate of $RO_2$ radicals. That said, the experimental results presented in this study only represent a barely-seen scenario in the atmosphere and does not have any atmospheric relevance. The authors are suggested to conduct 'low-$NO_x$' experiments as well where $H_2O_2$, for example, can be used as the OH precursor. The observed increase in SOA yields, together with the high-NO experiments, can be used as upper and lower limits for the a-pinene SOA aging in the atmosphere.*

Fundamentally, the issue of "atmospheric relevance" is one of whether the elementary chemical reactions occurring in the experimental system are the same as in the atmosphere. In this case, the question is the fate of peroxy radicals ($RO_2$). The chemical meaning of "high-$NO_x$" is "high-NO" in the sense that the dominant fate of $RO_2$ radicals is to react with NO. Therefore, by working at relatively high NO concentrations, we isolate the high-$NO_x$ pathway for the aging reactions. This is the same as experiments using $H_2O_2$ photolysis, which typically have extremely high $H_2O_2$ concentrations and thus a large flux through the reaction $OH + H_2O_2 \rightarrow H_2O + HO_2$. Those experiments tend to isolate the $RO_2 + HO_2$ pathway. In general, it is an error to conflate

"atmospherically relevant" with "atmospheric concentrations," though it is important to establish that the rate-limiting chemistry is relevant to the atmosphere. We have added the above important point in the revised paper.

**(3)** *The methodology of this study, including the chamber experimental approach, SOA yield measurements corrected by particle wall loss, O:C ratio calculations by AMS measurements, has been widely used in the community for decades. The main result of this study, i.e., SOA yields increase by 20-40 percent upon OH-oxidation aging, is insufficient to warranty publication of this manuscript. The authors should endeavor to explore more detailed mechanisms involved in the a-pinene SOA aging by thoroughly analyzing the AMS data, e.g., What functionalities/products have changed and contributed to SOA production during aging? Any indication of PMF analysis, for example, on the key processes involved in the chemical aging?*

PMF analysis works by separating the AMS-measured mass spectra into individual factors, and comparing those factors to existing libraries of mass spectra for the purpose of identifying potential sources for the SOA components. It is mostly used on ambient measurements where the sources for the OA are uncertain, while in these chamber experiments, we know that the SOA is formed via monoterpene oxidation. Application of this technique usually results in two or three factors with the fresh, aged and may be an intermediate product. While there could be useful information in this analysis it rarely provides the necessary chemical insights in cases where the changes in the spectra are relatively small like here.

In an attempt to explore the functionalities/products that may have changed during aging, we used the AMS high-resolution family analysis. Each fitted ion is grouped into a "family" based on their chemical formula, and the families used are: CH, CHO, $CHO_2$, $C_x$, HO, and NO. These are the main components of the organics formed, with family HO calculated by subtracting the other families from the total organic signals, because sulfate can fragment into water and thus interfere with family HO. We used family NO

to represent the organonitrate compounds formed during the aging step.

Based on the results, for example, of Experiments 1 and 2, the less oxidized ion family CH decreased around 10 percent during the aging process (from 41.9 to 38.1 percent of the OA in Exp. 1 and from 40.5 to 35.3 percent in Exp. 2) while the more oxidized components $CHO_2$ increased 4 percent in Exp. 1 (from 12.8 to 13.3 percent) and 16 percent in Exp. 2 (from 14.9 to 17.3 percent). The changes in the family CHO were +4 percent in Exp. 1 and -6 percent in Exp. 2 suggesting that there is both production and destruction of the corresponding family members. The organonitrate compounds as expected were close to zero initially in these experiments. In the end of the aging process the NO family represent 3-3.5

$CO_2^+$ (*m/z* 44) from family $CHO_2$ and $C_2H_3O^+$ (*m/z* 43) from family CHO are usually identified in aged and relatively fresh aerosols, respectively. Their fractions of the total organics, $f_{44}$ and $f_{43}$, have been used as chemical indicators in chamber experiments (Donahue et al., 2012). During the dark ozonolysis period of Exp. 1, the $f_{43}$ increased initially and stayed practically constant after t=0.2 h while $f_{44}$ decreased. This indicates the majority of the SOA formed initially was fresh. After the first introduction of OH, both $f_{43}$ and $f_{44}$ showed a stepwise increase, indicating formation of relatively fresh SOA and oxidation of the SOA. After the second introduction of OH, the $f_{43}$ decreased while $f_{44}$ increased over time until the end of the experiment, indicating the formed SOA was getting more oxidized with aging. During Exp. 2, the $f_{43}$ increased sharply initially and then slowly decreased over the dark ozonolysis period. This is consistent with the "ripening" effect observed during the MUCHACHAS campaign (Donahue et al., 2012). Overall, $f_{43}$ decreased while $f_{44}$ increased over the course of Exp. 2, indicating formed SOA being oxidized during aging.

We added the above analysis to the revised paper and also added the corresponding figures to the Supplementary Information.

*Specific:*

**(4)** *Page 2, Line 35-36: There have been a number of studies investigating the SOA aging processes using both static chamber and flow tube reactors (e.g., Robinson et al., Science, 2007; Loza et al., ACP, 2012; Lambe et al., EST, 2012). The authors need to revise the statement 'Most laboratory studies of SOA formation so far have focused on the ...'*

We have revised the corresponding sentence noting that the early SOA studies focused on the first stage of reactions.

**(5)** *Page 3, Line 64: References need to be given here.*

We have the corresponding three references at this point.

**(6)** *Page 6, Line 154: What is the particle size range measured by SMPS?*

The SMPS was set to measure particles from 15-700 nm for the experiments in this work. This information has been added to the paper.

**(7)** *Page 6, Line 171: The OH radical molar yield from ozonolysis of a-pinene is roughly 0.7. So technically, the measured SOA yield at first stage is already a result of certain OH aging. Have the authors estimated the fraction of a-pinene that was oxidized by OH radical in the ozonolysis experiments? Have the authors considered adding any OH scavengers (e.g., $H_2O_2$, CO, methanol) in the first stage of the experiments?*

This is a good point. OH scavengers cannot be used in the first phase of these aging experiments, because they will react with the OH in the second phase and will not allow the chemical aging to take place. The exception is hydrogen peroxide, as described in Henry et al. (2011), which acts as an OH scavenger (producing $HO_2$) under dark conditions but as an OH source under UV illumination. However, a-pinene is much more reactive (by a factor of 3-10) than the first-generation products and so most of the OH radicals react directly with a-pinene precursor. Because we do not employ a

scavenger, the first phase of the experiment includes reactions of the precursor and to a limited extent of the first-generation products with OH. We have used a kinetic box model of the system to estimate the extent of these reactions. Assuming a molar yield of 0.7 for a-pinene ozonolysis the model predicts that a little more than one third of the a-pinene reacted with OH in these experiments. Please note that these two reactions also take place together in the ambient atmosphere. This information has been added to the revised manuscript.

**(8)***Page 7, Line 205: The assumption that the particle loss rate in the 300-600 nm range is the same potentially leads to underestimation of the corrected SOA yields, as bigger particles have higher wall loss rate due to gravity deposition.*

Gravitational settling predominantly affects particles bigger than 1 micrometer (Crump and Seinfeld, 1981; McMurry and Rader, 1985), while the majority of the particle mass in our experiments was in the range of 100-600 nm. We have regularly performed experiments with seed particles to characterize the k's for the aforementioned size range and the loss constant does not vary in our chamber from 300 to 600 nm. We evaluated the errors caused by using a constant fit value for particles of 300-600 nm by applying the same k's to the seed loss periods during Exp. 1. The variability for the final seed loss period (t=4.2-8.5h) was 4.2 percent, indicating the corrected seed particle volume stayed relatively constant (Fig. 3). We have assumed the same uncertainty of 4.2 percent for particles during the SOA formation. This explanation has been added to the paper.

**(9)** *Page 8, Line 236: The authors are suggested calculate the SOA density from the AMS measured O:C and H:C ratio of SOA (Kuwata et al., EST, 2012) and compare with the current value used.*

We used an SOA density of 1.4 g cm$^{-3}$ in our calculations based on results from previous studies. We also used the approach described in Kostenidou et al. (2007),

matching the AMS-mass/composition distribution with the SMPS measured volume distributions and estimated an average density of 1.3±0.15 g cm$^{-3}$. The Kuwata et al. (2012) parameterization also predicted 1.3 g cm$^{-3}$. This information has been added to the manuscript.

**(10)** *Page 9, Line 237-238: Adding the 'V(t)/Vmax' factor in the SOA yield correction procedure does not seem necessary. The authors stated that 'deviations of V(t) from Vmax are caused by the uncertainty associated in applying the size dependent wall loss corrections'. As we know, particle wall loss rate depends on a few parameters, including chamber size, eddy diffusion, static charges on the Teflon surface, and etc. For most experiments presented here that only lasted for a few hours, these parameters should be fairly constant so that the particle wall loss rate should be quite consistent before and after one experiment. Where are the deviations originating from? I wonder if the assumption that particle loss rate in the 300-600 nm range is the same has any impact on the SOA yield correction here.*

We defined the parameter, $\epsilon_V$ (Line 223), as an estimation for the uncertainties caused by our particle wall-loss correction. We calculated an $\epsilon_V$ of 4.2 percent for Exp. 1 based on how much the loss-corrected seed volume (t=4.2-8.5 h) deviated from its average value. The deviation of V(t) from Vmax is likely caused by the uncertainty of our particle wall-loss correction, which is 4.2 percent in this case, and may well include the effect of using a constant value for particles ranging in size of 300-600 nm. When calculating the corrected SOA volume, we have to subtract the seed from the corrected total volume concentration. With the goal in mind of minimizing the impact of the uncertainties caused by the particle loss correction on formed SOA, we introduced the 'V(t)/Vmax' factor to attribute part of this 4.2 percent error to the seed. If we subtract a constant seed volume, we are intrinsically attributing this 4.2 percent error all and only to the corrected SOA and we get the lower limit of the corrected SOA volume.

**(11)** *Page 27, Figure 5: Why the O:C ratio of SOA kept decreasing in the second dark*

*period? What are the concentrations of NO$_x$ and O$_3$ during this period? Is there any NO$_3$-initiated chemistry going on?*

Similar evolution of the AMS spectra during periods without photochemistry has been observed in a number of similar studies. Donahue et al. (2012) called this evolution ripening to distinguish it from OH aging. The nature of this process is not well understood but it probably involves heterogeneous reactions. In our experiments the ozone concentration during this period was practically zero, so the production of NO$_3$ and the consecutive reactions were highly unlikely. We have added a short discussion and the corresponding reference to ripening in the paper.